# Educational Model for Evaluation of Airport NIS Security for Safe and Sustainable Air Transport

**Miroslav Kelemen** [1] , **Volodymyr Polishchuk** [2,*] , **Beáta Gavurová** [3] , **Rudolf Andoga** [1] , **Stanislav Szabo** [1] , **Wenjiang Yang** [4] , **John Christodoulakis** [5] , **Martin Gera** [6] , **Jaroslaw Kozuba** [7] , **Peter Kaľavský** [1] **and Matej Antoško** [1]

1   Faculty of Aeronautics, Technical University of Kosice, 04121 Kosice, Slovakia;
    miroslav.kelemen@tuke.sk (M.K.); rudolf.andoga@tuke.sk (R.A.); stanislav.szabo@tuke.sk (S.S.);
    peter.kalavsky@tuke.sk (P.K.); matej.antosko@tuke.sk (M.A.)
2   Faculty of Information Technologies, Uzhhorod National University, 88000 Uzhhorod, Ukraine
3   Research and Innovation Centre Bioinformatics, USP TECHNICOM, FBERG, Technical University of Košice,
    04001 Kosice, Slovakia; beata.gavurova@tuke.sk
4   School of Astronautics, Beihang University, Haidian District, Beijing 100191, China;
    yangwjbuaa@buaa.edu.cn
5   Faculty of Physics, National Kapodistrian University of Athens, GR-15784 Athens, Greece;
    ichristo@phys.uoa.gr
6   Faculty of Mathematics, Physics and Informatics, Comenius University in Bratislava, Bratislava,
    84248 Mlynská Dolina, Slovakia; mgera@fmph.uniba.sk
7   Faculty of Transport, Silesian University of Technology, 44100 Gliwice, Poland; jaroslaw.kozuba@polsl.pl
*   Correspondence: volodymyr.polishchuk@uzhnu.edu.ua; Tel.: +380-664207484

**Abstract:** One of the praxeological problems of safe and sustainable air transport (airfreight transport/air cargo, and air passenger transport) is the prevention and management of risks by competent staff, with the support of modern information and communication technologies. This paper presents an educational information model and software for the airport network and information systems risk assessment, primarily intended for aviation education and training of professionals for ensuring safe and sustainable air transport. The solution to the problem is based on the application of the fuzzy logic method in the air transport environment. Based on a fuzzy expert model, the selected scenario, and the input data established separately for airport assets by a group of 23 experts from aviation practice and a university, the following three assessments of airport network information system assets were constructed: Asset $A_2$ (meteorological information systems) has an insignificant risk with an estimated 0.1162, and assets $A_1$ (air traffic control and management (ATM), navigational aids and approach) and $A_3$ (runway monitoring system) received a low risk of airport network and information systems (NIS) security with ratings of 0.2623 and 0.2915, respectively. An airport NIS risk assessment was aggregated (0.2288), indicating a low degree of security risk to the airport's network and information systems. The aggregated risk assessment of airport NIS, including financial loss data, was calculated as 0.1438, representing a low degree of security risk to the airport's network and information systems. Scenarios for evaluating airport assets are changing for students during education. The results of the developed model and its software will be part of the Simulation Center of the Faculty of Aeronautics.

**Keywords:** airfreight transport; analysis of systems; processes and algorithms; sustainability; risk assessment; network and information systems (NIS); incident rate; airport assets

---

## 1. Introduction

In the context of the crisis caused by the COVID-19 pandemic and its consequences, the crucial importance of the sustainable transport of goods in the coordinated activities of international supply chain management has been highlighted more than ever. Airfreight transport (air cargo) is one of the key elements of the global transport system, and an important part of air transport is air passenger transport. According to the International Civil Aviation Organization, today's aircraft move well over USD 5 trillion worth of goods each year by air. Worldwide, air passenger numbers with a small cargo continued to rise, reaching 4.5 billion journeys annual, creating challenges for air cargo and mail security and facilitation, so maintaining or improving all aspects of air cargo safety and air passengers transport security and safety is essential.

The paper presents an information model and software for airport network and information systems risk assessment, primarily intended for aviation education and training of professionals to fulfill such roles for ensuring safe and sustainable air transport.

The focus of the proposed methodology is based mainly on our national 65 years' experience in aviation education and practice with the support of international experience of partners. The Slovak Republic faces specific problems including a lack of multi- and interdisciplinary education of specialists for integrated personal protection, data protection, information protection, airport security, air transport security and safety, flight safety, crisis management, and crime prevention for safe and sustainable air transport. The quality of the training of aviation specialists in this area is important, especially given the potential for loss of life for attacks on aircraft operating systems, air traffic control systems, airport network and information systems (NIS), etc. When we consider the sustainability of air transport, security and safety are at the forefront. Lastly, managing the risks in the field of antibacterial and antiviral protection of persons and contamination of material surfaces in air transport is a challenge, not only during this pandemic period, but also for everyday aviation practice. Finding solutions to the praxeological problem of educating these specialists with the support of current Software (SW) tools remains our challenge.

What is novel in the presented approach? The idea of using fuzzy logic in the security models is not new. The paper expands upon the applications of fuzzy logic theory to the digital space of risk assessment of information systems and data protection within airports and air transport, which it also offers within the simulation center and internationalization of these processes as part of the digital aviation education of aviation specialists. Traditionally, experts are included in the risk assessment strategy and valuation of different aspects of security and safety risks, which is common practice. However, we lack the tools for the multi- and interdisciplinary education of students, the new and future specialists who must be competent and skilled for incorporation into teams for risk assessment strategies and validation of various aspects of security and safety risks in aviation practice. To fulfill these roles in the future, staff need to be selected, educated, and trained, and as members of a multi- and interdisciplinary team of aviation risk assessment experts they must be prepared within multi- and interdisciplinary teams of students from the beginning of their professional careers. The results of the study provide practical tools to support these educational processes of air transport personnel.

The plan was to compare at least three relevant methods with metrics for the objective and comprehensive assessment of the risks and incidents of airport network and information systems (NIS) in the air transport sector. The first part of the study focused on innovative solutions used by fuzzy methods and models for civil airport NIS risk assessment. This paper presents the risk assessment in the first part of the ongoing study. In the second part of the study, we used a fuzzy multi-criterion decision-making method (FMCMD) that we developed. In the third part of the study, we used a selected expert system (rule-based expert system). The knowledge gained in individual research questions allowed the relevant methods for these purposes to be compared and practical conclusions to be drawn. We intend to use these study conclusions for a complex solution for the education at the Simulation Center of the Faculty of Aviation of the Technical University in Košice, Slovak Republic,

for the training of aviation specialists in this agenda, and to support the development of methodologies for the creation of intelligent systems for measuring and assessing security aspects of civil airport NIS.

In any intelligent decision support systems, a final decision maker (DM) is required. These systems should help, advise, and raise the level of validity of decision-making. Therefore, the decision support is improved with the formalization of expertise together with the quantitative assessments of various aspects of the assessed object. Air transport businesses need to complement the existing solutions with components responsible for security analysis, attack detection, and risk management in cybercrime prevention to align the information security systems with the current requirements [1,2]. Our research focused on developing risk management technologies with the involvement of NIS expertise in an adaptive approach to airport network security for civil aviation, primarily for air transport.

*Overview of Domestic and Foreign Research Studies*

The urgency of this task is demonstrated by major global studies and scientific publications assessing the risks of NIS security against various threats related to cybercrime prevention.

Simulation technologies and modeling techniques effectively support the assessment of cyber security risks in aviation [3]. To achieve this, it is recommended to develop, maintain, and apply simulation models and cyber-attack modeling scenarios to connect and develop simulation models from gate to gate, and integrate human interaction with cyber-attack modeling scenarios. Thorough evaluations and reviews of cyber risks for companies with internet access have already been explored, such as a recent study presenting a cyber risk vulnerability management software platform for simplifying and improving automation and continuity in cybersecurity assessment [4], also offering risk assessment and classification to the user. Bayesian networks, with various types of induced uncertainty with simplicity of criteria, have also been successfully used in the construction of an expert model for the analysis of cyber threats [5]. This means a fuzzy set of theories should be used to reflect knowledge of an object [6].

To properly assess the risk of airport NIS, it is necessary to learn how to scientifically model information uncertainty by drawing the formally described boundaries between reliable knowledge, knowledge with a certain level of reliability, and what is unknown [7]. To do this, the fuzzy-plural descriptions are used to model uncertainty [8,9]. For example, the fuzzy-plural description works on current information systems and technology [10] or artificial intelligence systems and decision support [11] on fixed point theorems in fuzzy metric spaces [12] or expert information using fuzzy logic [13], considering the general ideas and benefits underlying the current views on the use of fuzzy logic in the decision support systems, as Skorupski J. and Uchroński P. [14] presented a model fuzzy logic model supporting the management of a security screening checkpoint's organization at an airport, but only the role of the human factor was considered.

Cyber-attacks and incidents cause financial losses, but obtaining this business information is difficult. For example, Zurich Insurance Company in the U.K. reported that nearly 850,000 small- and medium-sized businesses were being hit by cyber-attacks, with more than one-fifth of the companies suffering losses of more than USD 13,000 and 1 in 10 companies losing more than USD 69,000 [15]. The 2017 Ponemon data leak damage study found that the average total cost for large companies was approximately USD 3.62 million. According to the findings of the study on global costs related to data leaks, the average cost of cyber-attacks more than doubled between 2014 and 2015, while the average cost of lost or stolen records increased slightly to nearly USD 41 [16]. In the 2018 study, the average cost of a data breach per compromised record was USD 148, and it took an average of 196 days for organizations to detect breaches. The 2019 report found that the average total cost of a breach ranges from USD 2.2 million for incidents with less than 10,000 compromised records to USD 6.9 million for incidents with more than 50,000 compromised records [17]. The 2018 study covered 477 organizations, including data on mega breaches for the first time. Although the cost per record remained consistent at USD 148, large-scale breaches of 50 million records could cost companies over USD 350 million [18]. As DeBrusk Ch. and Mee P. [19] correspondingly reported, the amounts spent by companies in an

attempt to protect themselves is also large, estimated at approaching USD 1 trillion annually on a global basis by 2022.

Today, no single method is available for creating risk management technologies by involving expertise using an adaptive approach. An expert model for assessing the risks and incidents of airport NIS using fuzzy sets is an urgently required task to improve civil aviation information security.

This paper is organized as follows. In Section 2, we describe the formal problem statement and the model of input data for assessment of the situation provided by a group of experts based on their knowledge, skills, and at least 15 years of practical experience. We present the fuzzy expert model for incident and risk assessment of airport NIS security for civil aviation. In Section 3, we outline the simulation experiment. In Section 4, we discuss the results of the expert model and its SW developed in the study. In Section 5, we conclude the paper and present the main results. We expand the ideas for future work and improvements.

## 2. Model of Experts' Input Data for Selected Airport Assets Assessment Scenarios

### 2.1. Formal Problem Statement, and Model of Input Data

Suppose we have a set of information assets of civil aviation security [2] $A = \{A_1; A_2; \ldots; A_n\}$, for which many threats to the security of personal data of network and information systems have been identified (the airport network threats in real-time, malicious or anomalous patterns, airport security threats, air transport security and safety threats, aviation communication systems threats, air traffic control threats, the threat of fraudulent acquisition and use of air passengers' private identification information, airport health threats including antibacterial and antiviral protection, etc.) $K = \{K_{i1}; K_{i2}; \ldots; K_{im}\}$, $i = \overline{1, n}$. Every security threat $K_{ij}$ for an asset $A_i$, $\left(i = \overline{1, n}, j = \overline{1, m}\right)$ is assessed by a group of experts in the form of the input data $\left(T_{ij}; \mu_{ij}\right)$, $\left(i = \overline{1, n}; j = \overline{1, m}\right)$. The input data have the following meanings:

- $T$ represents the consequences of threat implementation $K$ of airport NIS personal data security. This indicator is defined as the value of the next term set: $T = \left\{ M; \quad A; \quad H; \quad C \right\}$, where $M$ is the minimal consequences of the threat, $A$ is the average consequences of the threat, $H$ is the maximum consequences of the threat, and $C$ is the critical consequences of the threat.

- $\mu$ is the degree of possibility of occurrence of an NIS airport threat. This parameter belongs to the interval [0; 1], with different values having the following content: 0 indicates it is impossible that the threat will occur, 0.4 indicates a minimal possibility of the occurrence of the threat, 0.6 indicates average threat occurrence possibility, 0.8 indicates high threat occurrence possibility, and 1 indicates critical occurrence of the threat.

For each asset, we have $L$, which indicates the severity of the incident's consequences for the asset. This indicator is defined as the value of the next term set: $L = \left\{ V; \quad C; \quad U \right\}$, where $V$ is vital (if its failure lasts more than 2 h as too many flights will be delayed immediately and then canceled), $C$ is critical (if its failure cannot last more than 24 h because too many flights will be delayed), and $U$ is useful or not applicable [2].

Based on the input of expert data, we assessed the security risk of personal data of airport NIS $R = \{R_1; R_2; \ldots; R_n\}$ separately for assets, calculated financial loss incidents (probability of occurrence of risk) for assets $Z = \{Z_1; Z_2; \ldots; Z_n\}$, and constructed one aggregate estimate $Y \in [0; 1]$ to make additional decisions to prevent cybercrime. The solution to this problem can be demonstrated in the form of a structural diagram of the expert model for evaluation of the civil airport NIS security to support security management with ensuring safe and sustainable air transport (Figure 1).

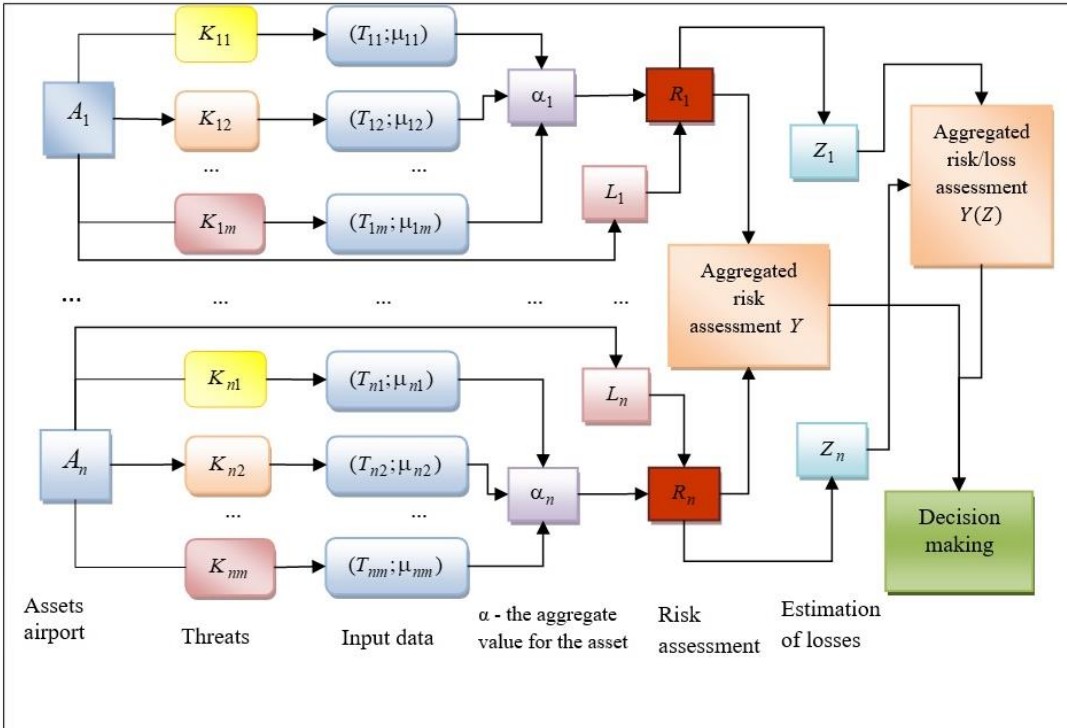

**Figure 1.** Structural diagram of the airport assets risk assessment expert model. $A_1$; $A_2$; …; $A_n$—information assets of civil aviation security; security threat for an asset; $(T_{ij}; \mu_{ij})$—the input data; $\alpha$—the aggregate value for the asset; $R_1$; $R_2$; …; $R_n$—risk assessment; $Z_1$; $Z_2$; …; $Z_n$—estimation of losses; $Y$—aggregated risk assessment; $Y(Z)$—aggregated risk assessment considering financial losses.

Assessment of situations using computer technology is knowledge-based (knowledge base and practical experience), and software solutions are used for the interpretation of rules for evaluation. The application of the peer-review method is largely based on the use of experienced experts who work on the basis of personal practice, intuition, and critical thinking and are able to anticipate, summarize, and apply their knowledge. The situation is classified according to the subjective judgments of experts (managers). The selected situation for the training of students determines the training scenario for specific lessons. Therefore, inputs on threats to airport NIS personal data were provided by a group of experts based on their knowledge, skills, practical experience, and additional sources, including:

- Eleven lecturers from aviation practice (air transport, security and safety, aviation communication systems, air traffic management) and airport management;
- Researchers' reports on identified vulnerabilities that could be used to address the threats;
- Computer investigators´ forensic reports of actual occurrences that analyzed the current situation in the field of computer security of civil aviation;
- Various studies and publications in the media related to computer crimes;
- Databases or directories listing and classifying threats.

Table 1 summarizes the NIS assets of the airport for civil aviation that we considered vital. We also present the severity of the effects of the asset incident [2] using the values of *L*.

**Table 1.** Airport network and information systems (NIS) assets. The severity of the effects of the asset incident.

| Assets | | The Name of the Asset | The Severity Consequences L |
|---|---|---|---|
| Safety and Security | $A_1$ | Access control systems | V (Vital) |
| | $A_2$ | Authentication systems | V |
| | $A_3$ | Baggage screening systems | V |
| | $A_4$ | Smart surveillance systems, e.g., smart CCTV, camera, and video | V |
| | $A_5$ | Customs and immigration | V |
| | $A_6$ | In-line explosive detection systems (IEDS) | V |
| | $A_7$ | Perimeter intrusion detection systems (PIDS) | V |
| | $A_8$ | Emergency response system | V |
| Management, Airline/Airside Operations | $A_9$ | Air traffic control and management (ATM), navigational aids and approach | V |
| | $A_{10}$ | Meteorological information systems | C (Critical) |
| | $A_{11}$ | Airport operation plan (AOP) | V |
| | $A_{12}$ | Network operation plan (NOP) | C |
| | $A_{13}$ | Departure control systems (DCS) | C |
| | $A_{14}$ | De-icing Systems | C |
| | $A_{15}$ | Airfield lighting control systems | V |
| | $A_{16}$ | Runway monitoring system | V |
| | $A_{17}$ | Communication, navigation, and surveillance (CNS) | V |
| IT and Computers | $A_{18}$ | Local area network systems (LAN) | V |
| | $A_{19}$ | Communications systems, e.g., radio spectrum management systems | V |
| | $A_{20}$ | Equipment hardware and software | C |
| | $A_{21}$ | Wide area network (WAN) | C |
| | $A_{22}$ | Network security management | C |
| Passenger Management | $A_{23}$ | Flight information display system (FIDS) | C |
| | $A_{24}$ | Passenger check-in and boarding | C |
| | $A_{25}$ | Staff authentication system, e.g., biometric identification system | C |

Consider the security threats to NIS personal data implemented using inter-network protocols [20] for airport assets.

$K_{i1}$ is the network traffic analysis. This threat is realized with the help of a special package parser program, intercepts all packets transmitted by the network segment, and selects from among them those in which the users´ ID and their passwords are transmitted. During the implementation of the threat, the intruder studies the logic of the network, that is, seeks to clear events that occur in the system.

$K_{i2}$ is network scanning. The essence of the process of the threat implementation is sending requests to the network services of the personal information system and analyzing their responses. The purpose is identifying protocols used, available network service ports, laws for the formation of connection IDs, the definition of active network services, the selection of user IDs, and passwords.

$K_{i3}$ is the threat of password detection. The purpose of this aspect is to gain unauthorized access by overcoming password protection. An attacker can implement the threat by a simple search, searching using special dictionaries, installing a malicious program to intercept the password, and substituting a trusted network object and intercepting packets. Threats are mainly used by special programs that attempt to gain access by sequential password selection.

$K_{i4}$ is the substitution of a trusted network object and transmission of messages on its behalf by granting its access rights. This threat is effectively implemented in systems that use unstable algorithms for user identification and authentication. Implementing a threat of this type requires overcoming the message identification and authentication system.

$K_{i5}$ is imposing the wrong network route. The possibility of imposing an erroneous route occurs due to the disadvantages inherent in routing algorithms, as a result of which the attacker can access

the network, where they can enter the operating environment of the technical tool in the personal information system.

$K_{i6}$ is the implementation of a false network object to intercept and violate a search query that will lead to the necessary change of route address data.

$K_{i7}$ is denial of service. These threats are based on network software weaknesses and vulnerabilities that allow an intruder to create conditions that, when detected by the operating system, prevents it from processing packages.

$K_{i8}$ is remote application launch. The threat involves running various pre-installed malicious programs on a personal information system server: bookmarks, viruses, and network spies. The main objective is to violate confidentiality, integrity, and availability of information, and obtain complete control. It also involves the unauthorized launch of user applications for the unauthorized reception of the necessary data breaches for launching application-driven processes.

For educational purposes, we have the airport facilities divided into four main areas and 25 subsystems of airport facilities ($A_1$–$A_{25}$), for which we can use (simulate) variable threats ($K_{ii}$) to the security of personal data in airport network and information systems, etc., which have the potential to have serious consequences. Our students should consider (identify, analyze, and evaluate) security threats to the airport NIS implemented using inter-network protocols for selected airport facilities, according to the situation formulated by their training instructors. The selected situation for the training of students determines the training scenario for specific lessons. The situation is classified according to the subjective judgments of experts (managers). We have a set of information assets of civil aviation security for which many threats to the security of network and information systems, etc., have been identified, and it is important to describe them to our students in some detail.

Thus, the input data of security threats on the assets of the airport, as received from the group of experts, are shown in Table 2.

**Table 2.** Experts' input data for airport assets threats assessment.

| Assets | Threats | Consequences of the Threat | Degree of Possibility of Occurrence of Threat | The Severity Consequences |
|--------|---------|----------------------------|-----------------------------------------------|---------------------------|
| $A_1$ | $K_{11}$<br>$K_{12}$<br>…<br>$K_{1m}$ | $T_{11}$<br>$T_{12}$<br>…<br>$T_{1m}$ | $\mu_{11}$<br>$\mu_{12}$<br>…<br>$\mu_{1m}$ | $L_1$ |
| … | … | … | … | … |
| $A_n$ | $K_{n1}$<br>$K_{n2}$<br>…<br>$K_{nm}$ | $T_{n1}$<br>$T_{n2}$<br>…<br>$T_{nm}$ | $\mu_{n1}$<br>$\mu_{n2}$<br>…<br>$\mu_{nm}$ | $L_n$ |

*2.2. Fuzzy Expert Model for Incident and Risk Assessment NIS Airport Security for Civil Aviation*

Here, we outline a model for conducting a comprehensive airport NIS risk assessment in the form of an algorithm consisting of the following steps:

Step 1. Fuzzification of the input data.

In the first step, we consider the dependence of the consequences of the implementation of the threat of airport NIS and the degree of possibility of realization. To do this, the following approach, based on fuzzy set theories, was used [21,22].

Consider the consequences of the threat of airport NIS. Their terms $T = \left\{ \; M; \quad A; \quad H; \quad C \; \right\}$ can be adequately determined on a percentage scale [0; 100], each of which is given an interval value [$a$; $b$]. Example, "$M$" = [0; 20], "$A$" = [20; 50], "$H$" = [50; 80], and "$C$" = [80; 100]. This approximation has the following meaning: If a threat is realized at, for example, 90%, then it is treated as a critical consequence of the threat.

The dependence of the consequences of threat realization $T_{ij}$ and the degree of possibility of such sales $\mu_{ij}$ $\left(i = \overline{1, \ n}; \ j = \overline{1, \ m}\right)$ are naturally considered as a membership functions statement "value $x$ greater". The higher the threat capability and the more critical the impact, the more dangerous the NIS incident, which entails a high security risk for airport operations. In this case, for example, we can take an S-like membership function or another similar to it [23,24]:

$$
\mu_{ij} = \begin{cases} 0, & x_{ij} \leq a; \\ 2\left(\frac{x_{ij}-a}{b-a}\right)^2, & a < x_{ij} \leq \frac{a+b}{2}; \\ 1 - 2\left(\frac{b-x_{ij}}{b-a}\right)^2, & \frac{a+b}{2} < x_{ij} < b; \\ 1, & x_{ij} \geq b. \end{cases} , \left(i = \overline{1, \ n}; \ j = \overline{1, \ m}\right). \tag{1}
$$

Because the probability of occurrence of the known threat is $\mu_{ij}$ and intervals of numerical values for $T_{ij}$ are known, then for each threat for the assets under consideration, we express our dependence $x_{ij}$ from Formula (1) as:

$$
x_{ij} = \begin{cases} \sqrt{\frac{\mu_{ij}}{2}}(b-a) + a, & 0 \leq \mu_{ij} \leq 0.5 \\ b - \sqrt{\frac{1-\mu_{ij}}{2}}(b-a), & 0.5 < \mu_{ij} \leq 1 \end{cases} \tag{2}
$$

where $a$ and $b$ are the values of the edges of the intervals, which depend on the linguistic variable $T_{ij}$, $\left(i = \overline{1, \ n}; \ j = \overline{1, \ m}\right)$. The terms are defined on a percentage scale [0; 100], then the obtained values $x_{ij} \in [0; 100]$ are normalized by:

$$
f_{ij} = \frac{x_{ij}}{100}, \left(i = \overline{1, \ n}; \ j = \overline{1, \ m}\right). \tag{3}
$$

The value obtained, $f_{ij}$, is named the incident rate, which is interpreted as a function of revealing the uncertainty of fuzzy expert judgment, the consequences of the implementation of the threat to the airport NIS, and the extent of its possible implementation on the $i$th asset of the $j$th threat. The greater the value $f_{ij} \in [0; \ 1]$, the greater the $j$th security threat to the airport NIS within the $i$th asset.

We present a graphical interpretation of the fuzzification of the input data in Figure 2.

So, in the first step, the incoming data were fuzzified and the NIS airport incident rate was obtained.

Step 2. Aggregation of the incident rate within the asset.

In the second step, we output the aggregate value, which we denote $\alpha_i$, for asset $A_i$ using the following formula:

$$
\alpha_i = \frac{1}{k_i} \sum_{j=1}^{m} f_{ij}, i = \overline{1, \ n}, \tag{4}
$$

where $k_i$ is the number of NIS airport threats to the asset $A_i$. Estimations $\alpha_i \in [0; 1]$ represent the quantitative normalized security indicators of the airport NIS of the relevant asset $A_i$. The greater the value $\alpha_i \in [0; 1]$, the worse the security of the airport NIS within the ith asset. So, when $\alpha_i = 0$, we evaluate 100% security of the $i$th asset.

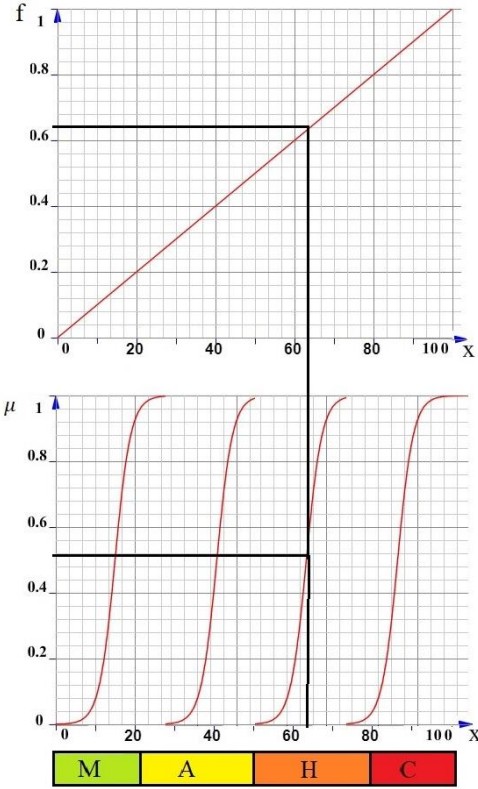

**Figure 2.** Fuzzification of the experts ' input data for the selected scenario of airport assets assessment. M is the minimum consequence of the threat, which is an interval value on the percentage scale [0; 20]; A is the average consequence of the threat, which is an interval value on the percentage scale [20; 50]; H is the high consequence of the threat, which is an interval value on the percentage scale [50; 80]; and C is the critical consequence of the threat, which is an interval value on the percentage scale [80; 100].

Step 3. Security risk assessment of the airport NIS.

NIS risk depends on the incident rate and severity of the consequences, *L*. Each value is evaluated by an expert or a team of experts at the intervals [0.1; 1], where V is Vital [0.8; 1], C is Critical [0.4; 0.8], and U is Useful or not applicable [0.1; 0.4].

We define the risk assessment as a multiplication by:

$$R_i = \alpha_i L_i, i = \overline{1, n}. \tag{5}$$

The greater the consequence for an airport asset, the greater the risk, and valuation *R* approaches one.

The estimation *R* has been normalized. Based on this, we determine the level of risk of the airport asset using the following linguistic interpretation $\{r_1, r_2, r_3, r_4, r_5\}$:

- $R_i \in [0; 0.1]$—$r_1$ is insignificant risk of airport NIS security by *i*th asset;
- $R_i \in (0.1; 0.3]$—$r_2$ is low risk of airport NIS security by *i*th asset;
- $R_i \in (0.3; 0.5]$—$r_3$ is average risk of airport NIS security by *i*th asset;
- $R_i \in (0.5; 0.7]$—$r_4$ is high risk of airport NIS security by *i*th asset;
- $R_i \in (0.7; 1]$—$r_5$ is the highest risk of airport NIS security by *i*th asset.

Step 4. Consideration of the importance of an airport asset.

Set weights for each asset for further decisions about the airport NIS security $\{w_1, w_2, \ldots, w_n\}$ from the interval [1; 10]. If the experts do not need the weights for each asset, without reducing the

generality, assets are considered in equilibrium. Because all the calculations are reduced to a single scale, we define the normalized weights separately:

$$v_i = \frac{w_i}{\sum_{i=1}^{n} w_i}, \ i = \overline{1, n}. \tag{6}$$

Step 5. Defuzzification of the data.

Next, we derive a summary of the airport NIS risk assessment for the assets under consideration for further decisions [25]. Estimates received $R_i$, $i = \overline{1, \ n}$ are normalized, but the focus of the goal must be changed because the smaller the $R$, the lower the risk, and the greater the risk, $v$, the more important the asset. To do this, we use one of the situations depending on the experts' opinion of the situation, changing the focus of the goal $R$. The final result is:

$$Y_1 = 1 - \frac{1}{\sum_{i=1}^{n} \frac{v_i}{(1-R_i)}} \text{ for pessimistic opinions of experts;} \tag{7}$$

$$Y_2 = 1 - \prod_{i=1}^{n} (1 - R_i)^{v_i} \text{ for careful expert opinions;} \tag{8}$$

$$Y_3 = 1 - \sum_{i=1}^{n} v_i (1 - R_i) \text{ for average expert opinions;} \tag{9}$$

$$Y_4 = 1 - \sqrt{\sum_{i=1}^{n} v_i (1 - R_i)^2} \text{ for optimistic expert opinions.} \tag{10}$$

Step 6. Calculation of incident financial loss for the assets.

The occurrence of security incidents for the airport NIS asset causes it to be inoperable, delayed, or canceled, resulting in financial loss. Asset work has a price for a certain period of time. The cost of rebuilding works depends on the asset. We calculate the financial losses separately for the assets $Z = \{Z_1; Z_2; \ldots; Z_n\}$, according to:

$$S_i = SL_i \cdot t_i + cr_i, \tag{11}$$

$$Z_i = S_i \cdot R_i, i = \overline{1, n}, \tag{12}$$

where $SL_i$ is the profit of the NIS (server) for one hour for the ith asset $A_i$, $t_i$ is the server recovery time, $cr_i$ is the the cost of renewal of NIS operation for the ith asset, $S_i$ is the total amount of financial loss in case of failure of airport NIS for the realized threat and the cost of reconstruction work, and $Z_i$ is the amount of the provision for financial losses in the implementation of risk.

Next, we perform informational modeling of the fuzzy representation of the amount of the reserve for financial losses using fuzzy sets apparatus. Because profit and loss are linear dependencies, this representation is naturally realizable in the form of a linear membership function:

$$\mu(Z_i) = \begin{cases} 0, & Z_i \leq a_i; \\ \frac{Z_i - a_i}{b_i - a_i}, & a_i < Z_i \leq b_i; i = \overline{1, \ n}. \\ 1, & Z_i > b_i. \end{cases} \tag{13}$$

The interval value is defined as the dependence of NIS (server) earnings per hour separately on assets: $a_i = S_i \cdot 0.1$, $b_i = S_i$, $i = \overline{1, \ n}$.

According to the resulting assessment $\mu(Z)$, we compare the inputs of financial losses in the implementation of risk. If $\mu(Z)$ equals 1, the risk has been realized and financial losses have been maximized. The built-in membership function (14) for the presentation of fuzzy knowledge allows considering the uncertain factors in decision-making and the experts uncertainty in their conclusions.

Step 7. Summary of financial incident data losses and their risks for airport assets.

We use a two-dimensional Gaussian function with the interval [0; 1] to aggregate the risks and financial losses of incidents according to [26]:

$$O_i = F\left(R_i; \ \mu(Z_i)\right) = e^{-\left((R_i-1)^2 + (R_i-1)(\mu(Z_i)-1) + (\mu(Z_i)-1)^2\right)}. \ i = \overline{1, \ n}. \tag{14}$$

Estimation $O$ take a value of one in the case of high risk and maximum possible financial losses.

Step 8. Calculations of airport NIS security risk assessment considering financial losses.

We perform a calculation similar to Step 5, but we use assessment $O$ instead of $R$.

$$Y_1\left(Z\right) = 1 - \frac{1}{\sum_{i=1}^{n} \frac{v_i}{(1-O_i)}} \text{pessimistic opinions of experts;} \tag{15}$$

$$Y_2\left(Z\right) = 1 - \prod_{i=1}^{n} \left(1 - O_i\right)^{v_i} \text{careful expert opinions;} \tag{16}$$

$$Y_3\left(Z\right) = 1 - \sum_{i=1}^{n} v_i\left(1 - O_i\right) \text{average expert opinions;} \tag{17}$$

$$Y_4\left(Z\right) = 1 - \sqrt{\sum_{i=1}^{n} v_i\left(1 - O_i\right)^2} \text{optimistic expert opinions.} \tag{18}$$

Estimates receiving $Y$ and $Y(Z)$ are normalized. Using the expertise of security experts, you can offer a linguistic representation as follows:

- If $Y/Y\ (Z)\ \in\ [0;\ 0.1]$—$y_1$ = small degree of security risk to the airport's network and information systems;
- If $Y/Y(Z)\ \in\ (0.1;\ 0.3]$—$y_2$ = low degree of security risk to the airport's network and information systems;
- If $Y/Y(Z)\ \in\ (0.3;\ 0.5]$—$y_3$ = average degree of security risk to the airport's network and information systems;
- If $Y/Y(Z)\ \in\ (0.5;\ 0.7]$—$y_4$ = high degree of security risk to the airport's network and information systems";
- If $Y/Y(Z)\ \in\ (0.7;\ 1]$—$y_5$ = maximum degree of security risk to the airport's network and information systems.

## 3. Results of Education Fuzzy Expert Model Verification According to Selected Student Training Scenario

In this section, we provide an example and verification of a fuzzy expert model for accident and security risk assessment for a civil airport NIS using the proposed algorithm according to the selected student training scenario. Training scenarios are created by training instructors from four main areas of airport facilities and their 25 subsystems. The number of areas and subsystems creates space for scenario variations. Specific scenarios are created first within one area and its several subsystems. Subsequently, the situation for students' decision-making is complicated by the creation of a scenario from at least two areas of airport facilities and four of their subsystems, etc. Each training scenario pursues an educational goal so that students acquire knowledge and skills in identifying, analyzing, and assessing risks in selected areas and subsystems of airport facilities, in creating solutions (strategies) and especially in risk management. Training scenarios are aimed at identifying and understanding the causes of failure of the human factor, the technical system, or a combination of these causes (resources).

We assessed the security risk of the airport NIS of the eight personal data security threats for three assets. Within the Simulation Center of the Faculty of Aeronautics of the Technical University in Košice, a group of experts simulated a situation based on which 11 lecturers from aviation practice and airport management, 8 teachers/instructors, and 4 Ph.D. students established the input data separately for asset (Tables 3–5).

**Table 3.** Input data expert assessment for the selected scenario of airport asset $A_1$

| Assets | Threats | Consequences of the Threat | Degree of Possibility of Threat Occurrence | The Severity Consequences |
|---|---|---|---|---|
| $A_1$ (air traffic control and management (ATM), navigational aids and approach) | $K_{11}$ | M (is the minimal consequences of the threat) | 0.4 | 0.9 |
| | $K_{12}$ | M | 0.6 | |
| | $K_{13}$ | A (is the average consequences of the threat) | 0.4 | |
| | $K_{14}$ | H (is the maximum consequences of the threat) | 0.8 | |
| | $K_{15}$ | A | 0 | |
| | $K_{16}$ | M | 0.8 | |
| | $K_{17}$ | M | 0.4 | |
| | $K_{18}$ | H | 0.6 | |

**Table 4.** Input data expert assessment for the selected scenario of airport asset $A_2$

| Asset | Threats | Consequences of the Threat | Degree of Possibility of Threat Occurrence | The Severity Consequences |
|---|---|---|---|---|
| $A_2$ (meteorological information systems) | $K_{21}$ | A | 0.4 | 0.6 |
| | $K_{22}$ | M | 0.4 | |
| | $K_{23}$ | A | 0.4 | |
| | $K_{24}$ | M | 0.8 | |
| | $K_{25}$ | A | 0 | |
| | $K_{26}$ | A | 0.6 | |
| | $K_{27}$ | M | 0.4 | |
| | $K_{28}$ | M | 0 | |

**Table 5.** Input data expert assessment for the selected scenario of airport asset $A_3$

| Asset | Threats | Consequences of the Threat | Degree of Possibility of Threat Occurrence | The Severity Consequences |
|---|---|---|---|---|
| $A_3$ (runway monitoring system) | $K_{31}$ | H | 0.8 | 0.8 |
| | $K_{32}$ | M | 0.4 | |
| | $K_{33}$ | M | 0.8 | |
| | $K_{34}$ | M | 0.6 | |
| | $K_{35}$ | A | 0.4 | |
| | $K_{36}$ | A | 0.8 | |
| | $K_{37}$ | H | 0.4 | |
| | $K_{38}$ | H | 0 | |

The calculation of aggregate estimates $Y$ and $Y$ $(Z)$, performed according to the algorithm developed above, is as follows:

Step 1. Fuzzification of the input data.

Calculate the value $x_{ij}$ and incident rate $f_{ij}$ using Formulas (2) and (3), respectively, and the result is shown in Table 6.

**Table 6.** Fuzzification of the experts' input data for the selected scenario of airport assets assessment.

| Threats | Values $x$ | Normalized Values $f$ | Threats | Values $x$ | Normalized Values $f$ | Threats | Values $x$ | Normalized Values $f$ |
|---|---|---|---|---|---|---|---|---|
| $K_{11}$ | 8.94 | 0.0894 | $K_{21}$ | 33.42 | 0.3342 | $K_{31}$ | 70.51 | 0.7051 |
| $K_{12}$ | 11.06 | 0.1106 | $K_{22}$ | 8.94 | 0.0894 | $K_{32}$ | 8.94 | 0.0894 |
| $K_{13}$ | 33.42 | 0.3342 | $K_{23}$ | 33.42 | 0.3342 | $K_{33}$ | 13.68 | 0.1368 |
| $K_{14}$ | 70.51 | 0.7051 | $K_{24}$ | 13.68 | 0.1368 | $K_{34}$ | 11.06 | 0.1106 |
| $K_{15}$ | 20 | 0.2 | $K_{25}$ | 20 | 0.2 | $K_{35}$ | 33.42 | 0.3342 |
| $K_{16}$ | 13.68 | 0.1368 | $K_{26}$ | 36.58 | 0.3658 | $K_{36}$ | 40.51 | 0.4051 |
| $K_{17}$ | 8.94 | 0.0894 | $K_{27}$ | 8.94 | 0.0894 | $K_{37}$ | 63.42 | 0.6342 |
| $K_{18}$ | 66.58 | 0.6658 | $K_{28}$ | 0 | 0 | $K_{38}$ | 50 | 0.5 |

Step 2. Aggregation of the incident rate within the asset.

In the second step, we output the aggregated values for the assets $A_1$, $A_2$, and $A_3$ using Formula (4): $\alpha_1 = \frac{1}{8}(0.0894 + 0.1106 + \ldots + 0.6658) = 0.2914; \alpha_2 = 0.1937; \alpha_3 = 0.3644$.

Step 3. Security risk assessment of NIS.

NIS risk depends on the incident rate and the severity of consequences $L$, which is determined by Formula (3): $R_1 = \alpha_1 \cdot L_1 = 0.2914 \cdot 0.9 = 0.2623; R_2 = 0.1937 \cdot 0.6 = 0.1162; R_3 = 0.3644 \cdot 0.8 = 0.2915$.

On the basis of the obtained experts' assessment, we determined the degree of risk of airport assets.

$R_1 \in (0.1; 0.3]$—$r_2$ is low risk of airport NIS security by asset $A_1$ (air traffic control and management (ATM), navigational aids and approach);

$R_2 \in [0; 0.1]$—$r_1$ is insignificant risk of airport NIS security by asset $A_2$ (meteorological information systems);

$R_3 \in (0.1; 0.3]$—$r_2$ is low risk of airport NIS security by asset $A_3$ (runway monitoring system).

Step 4. Consideration of the importance of an airport asset.

Let the security experts set the weights accordingly for each asset {10; 8; 9}, then we define the normalized weights for Formula (6): $v_1 = 0.37; v_2 = 0.3; v_3 = 0.33$.

Step 5. Defuzzification of the data.

We derive an aggregate airport NIS risk assessment for the assets under consideration and use a convolution, for example, the average expert considerations, for Formula (9):

$$Y_3 = 1 - (0.37(1 - 0.2623) + 0.3(1 - 0.1162) + 0.33(1 - 0.2915)) = 0.2288.$$

Step 6. Calculation of financial loss incident for assets.

Let the security experts have information for the assets provided in Table 7.

**Table 7.** Input data for the calculation of financial losses.

| Assets | $SL$—Profit of NIS (Server) Per Hour (USD) | $t$—Recovery Time Server Operation (h) | $cr$—Cost of Recovery Work of NIS (USD) |
|---|---|---|---|
| $A_1$ | 25,000 | 3 | 5000 |
| $A_2$ | 15,000 | 2 | 2000 |
| $A_3$ | 20,000 | 5 | 8000 |

Next, we calculated the total amount of financial losses using Formula (11), the amount of the provision for financial losses at risk using Formula (12), and $\mu(Z)$ using Formula (14). The result is shown in Table 8.

Step 7. Aggregate the financial loss due to data loss and their risks for the airport assets using Formula (14): $O_1 = 0.1619; O_2 = 0.0733; O_3 = 0.1865$.

Step 8. Calculate the airport NIS security risk assessment considering financial losses.

This calculation is feasible according to the average expert opinion, using Formula (17):

$$Y_3(Z) = 1 - (0.37(1 - 0.1619) + 0.3(1 - 0.0733) + 0.33(1 - 0.1865)) = 0.1438.$$

**Table 8.** Calculation of incident financial loss for the selected scenario of airport assets assessment.

| Assets | S | Z | $\mu(Z)$ |
|---|---|---|---|
| $A_1$ | 80,000 | 20,981.7 | 0.180301 |
| $A_2$ | 32,000 | 3719.52 | 0.018039 |
| $A_3$ | 108,000 | 31,486.32 | 0.212822 |

Therefore, based on the fuzzy expert model, incident assessment, and the security risk of the airport NIS for civil aviation, the following three assets were obtained: Asset $A_2$ (meteorological information systems) has an insignificant risk with an estimated $R_2$ of 0.1162. Assets $A_1$ (air traffic control and management (ATM), navigational aids and approach) and $A_3$ (runway monitoring system) received a low risk of airport NIS security with ratings $R_1 = 0.2623$ and $R_3 = 0.2915$, respectively.

An airport NIS risk assessment was aggregated at $Y = 0.2288$, indicating a low degree of security risk to the airport's network and information systems.

Aggregated risk assessment of airport NIS, including financial loss data, was calculated as $Y(Z) = 0.1438$, representing a low degree of security risk to the airport's network and information systems.

Further management decisions are made by the airport NIS experts based on aggregated estimates within the intelligent decision support systems for the final airport security decision-maker.

As part of the research, we developed innovative software that we have named the Expert Model for Evaluation of the Civil Airport NIS Security based on the constructed model and algorithmic support (Figure 3).

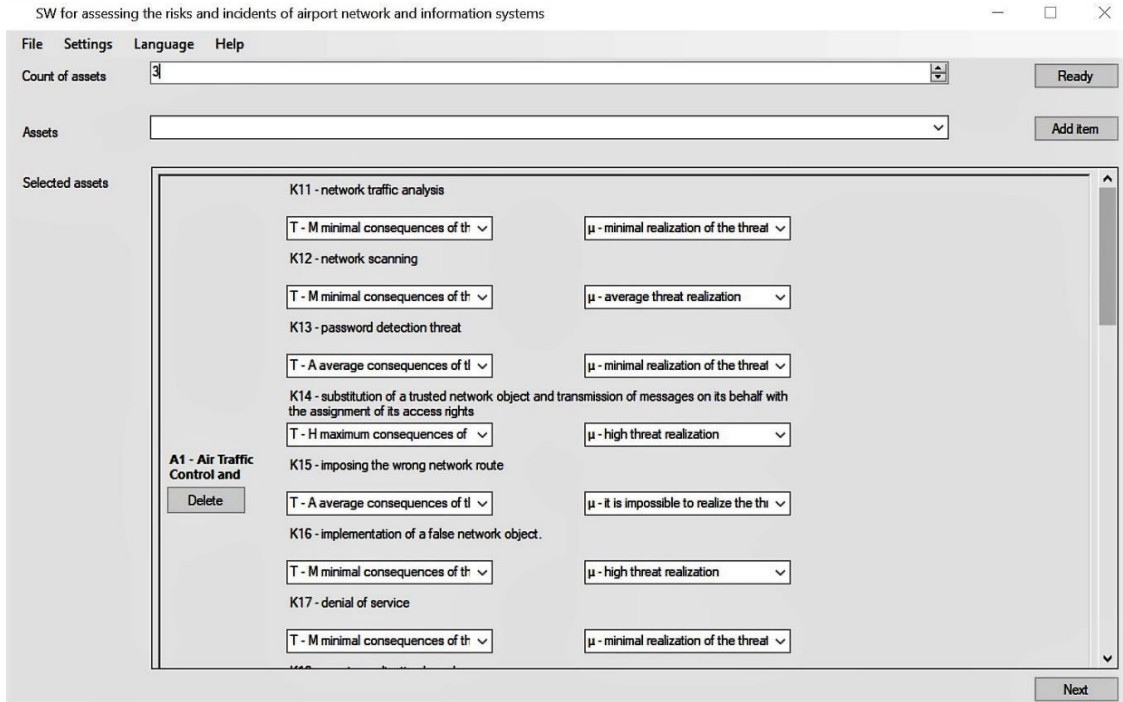

**Figure 3.** Software window for the modular system of the selected airport assets' assessment according to the students training scenario ($A_1$, Air Traffic Control and Management).

The software was designed for the Simulation Center of the Faculty of Aeronautics at the Technical University in Košice for the training of aviation specialists, which is open to cooperation with other institutions, and the software can be adapted for other users too. The software will extend the digital aviation education tools and the current approach to the training of aviation professionals, while having the potential to strengthen internationalization in aviation education.

All important components upon which the proposed algorithm works were placed in the software settings (Figure 4). After entering all the input data on the assets is a calculation for the proposed algorithm. Step 8 presents an aggregate assessment of the airport's security risk, taking into account financial losses, on the basis of which further management decisions are made (Figure 5).

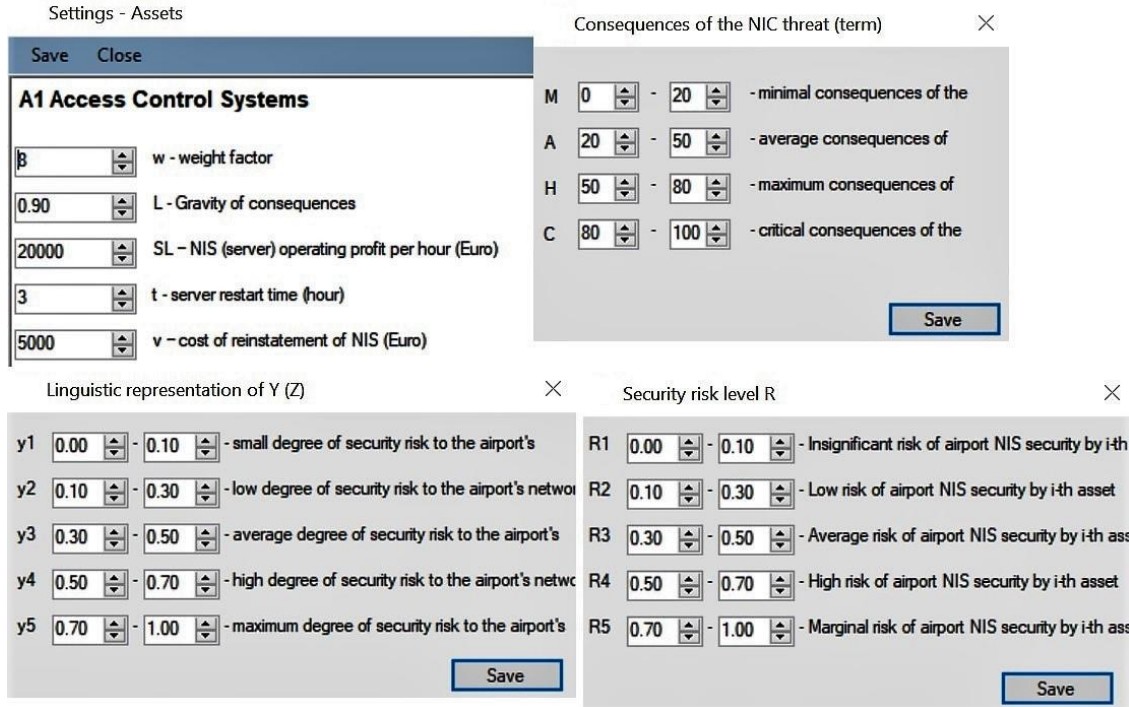

**Figure 4.** Software settings for consequences, linguistics, and numerical security risk level for airport network information system assessment according to the student training scenario.

**Figure 5.** Screenshot of the final assessment of selected airport assets NIS risk for support of decision-makers (students).

The working results of the created software obtained:

- Setting up of 25 airport NIS assets and eight personal data security threats to assess the airport NIS security risks. The input data are presented in the form of linguistic assessment (term-set of four linguistic variables) of consequences of the realization of security threats of personal data of NIS of the airport and degree of potential threat occurrence;
- An approach for calculating the number of financial loss reserves for the realization of airport NIS risk is presented and a function for interpreting the dependence of financial loss reserves is created;

- An education model for obtaining an aggregate airport NIS risk assessment is presented in the form of an eight-step knowledge-based algorithm, using linguistic variables, which reveals the inaccuracy of the input data, for preventing cybercrime and producing the outputs: risk assessment and linguistic interpretation of the airport NIS safety, based on the "incident rate" and the severity of the effects of the incident on the asset; aggregated assessment and linguistic interpretation of the airport NIS risk for assets under consideration concerning their importance; aggregated assessment and linguistic interpretation of airport NIS security risk taking into account financial losses.

## 4. Discussion

The expert model developed in this study, which can be used to assess the security risks of the airport NIS using the expert's analysis with their knowledge, reveals the inaccuracy of input assessments and increases the validity of further management decisions based on the results obtained. The output of the model is the risk measuring and assessment of airport NIS and its linguistic interpretation, the aggregated risk assessment of airport NIS for the assets under consideration, and aggregated risk assessment of airport NIS, considering the financial losses with the linguistic interpretations. Innovative software was developed for the practical implementation of the model for the air transport environment within aviation education at a simulation center.

The personnel within the air transport ecosystem work in different workplaces, in different work positions. Individual failure due to the human factor, incidents regarding work procedures, or technology failure in one section of airport information systems may jeopardize the integrity and security of the entire air traffic support system. For this reason, students are trained in three stages: general aviation education, training of aviation specialists, and training using virtual training and information, simulation, and communication technologies in the aviation environment. The proposed model enables the division of students, the future employees of airports, into the following basic groups: safety and security, management/airline/airport operations, information technology (IT) and computers in aviation, and passenger management and services (air cargo and air passenger transport).

As part of general aviation education providing the required aviation knowledge and skills of staff, students are prepared for the identification, analysis, assessment, and management of risks with an emphasis on:

(1) Safety and security: airport security, protection of persons and property, authentification systems, baggage screening systems, smart surveillance systems, customs and immigration services, in-line explosive detection systems (IEDS), perimeter intrusion detection systems (PIDS), and emergency response systems;

(2) Management/airline/airport operations: air traffic control and management (ATM), navigational aids and approach, meteorological information systems, airport operation planning (AOP), network operation planning (NOP), departure control systems (DCS), ground handling, de-icing systems, airfield lightning control systems, and runway monitoring systems;

(3) IT and computers in aviation: information systems for managers, local area network systems (LAN), communications systems, radio spectrum management systems, hardware and software at the airport/in the air transport, wide area network (WAN), network security management in the aviation environment, flight information display systems (FIDS), etc.;

(4) Passenger management and services (air cargo and air passenger transport): air cargo security procedures, passenger check-in and boarding, staff authentication system, etc.

The modular system of the proposed information model respects the mentioned areas of education and provides functionalities for the preparation of students for aviation according to various scenarios and input data from the expert database relevant to these areas in aviation. The proposed modular system forms the basis for the second level of education, the aviation specialists, in which its modules and functionalities are extended; for example, airport security and crisis management, decision support

for airport security, and crisis management for the protection of critical infrastructure entities in times of emergency such as COVID-19 for airport companies.

It was not possible to use all possible threat and risk scenarios in the study within the air transport environment for the aviation education of students. The proposed algorithm and created SW of the web application extend the currently available aviation education tools. The algorithm was verified on an example of an evaluation of three selected airport network assets: $A_1$ (air traffic control and management (ATM), navigational aids and approach), $A_2$ (meteorological information systems), and $A_3$ (runway monitoring systems), according to the selected scenarios and input data from 23 experts from aviation practice and aviation education.

An example from selected airport network assets allows students to build the ability to identify, describe, and analyze the threat and risks, as well as numerically express the level of risk. The proposed methodological procedure is applicable to other areas and workplaces in aviation. The model provides another tool for quantifying these phenomena, their complexity, and their causality, helping to increase knowledge in the field, including awareness of the importance of the human factor in all processes relevant to air traffic and air transport. For the next generation of students, the proposed model interestingly describes the issues in aviation that have the potential to cause material and financial damage, and even the loss of human lives.

Sharing research knowledge directly linked to real-world testing with a focus on creating a predictable platform in the field of cybersecurity education will support the current lack of public sector cooperation with the private and academic sectors. This will create an institutional framework for systematic, coordinated, and effective work, particularly at the strategic level.

In addition to the importance of sharing the research findings of international and national research teams and implementing the findings into action plans according to national strategies, the scientific outcomes can significantly impact the quality of cyber/information security education systems for air transport where discrepancies exist at different levels of competency and preparedness for cyber threats. We will not achieve modern and sustainable air transport without individual and institutional capabilities and resilience.

## 5. Conclusions

The World Economic Forum in 2020 [27] identified a list of global risks in many critical sectors. In terms of concerns, technology sector information, infrastructure failures, and cyberattacks are considered to be at the top of the global risks. Critical infrastructure providers, including the airport transport network and information systems, must take risk management measures as well as permanently monitor and report serious incidents, among other things. A cyber/information risk management system will not work without innovative and effective methodologies, models, and concepts, which are the products of research teams. This need for development and implementation into real life is increasing as the complexity of information systems increases and the impact of potential threats is reinforced, having potentially highly negative consequences for society, its protection, and security. Sustainable air transport is therefore linked to security and safety management and preventive measures to ensure an acceptable level of security and safety risks.

In this study, we developed an educational expert model of airport NIS risk and security incidents measuring and assessment in the framework of the aviation sector. The major outcomes, advantages, and limitations of the proposed work are as follows:

The major outcomes include:

- The creation of an educational expert model based on fuzzy logic (the modular system of the proposed educational information model) for obtaining an aggregate airport NIS risk assessment was presented in the form of an eight-step knowledge-based algorithm using linguistic variables, which reveals the inaccuracy of the input data for preventing cybercrime. The following outputs were produced: risk assessment and linguistic interpretation of the airport NIS safety based on the incident rate and the severity of the effects of the incident on the asset, aggregated assessment

and linguistic interpretation of the airport NIS risk for assets under consideration concerning their importance, and aggregated assessment and linguistic interpretation of airport NIS security risk considering financial losses;

- Based on the developed educational expert model, innovative software was constructed for the Simulation Center of the Faculty of Aeronautics at the Technical University in Košice, constructed for the multi- and interdisciplinary training of aviation specialists. The algorithm of the education information model was presented and verified on an example of airport NIS security risk measurement and assessment on eight personal data security threats and three airport NIS assets.

The advantages of our findings include:

- Increasing the objectivity of expert assessments using the input linguistic variables of consequences of the occurrence of security threats to personal information in airport NIS and the degree of possibility of such risks. The findings revealed the uncertainties in fuzzy expert judgment through the incident rate function; it performs the risk assessment and linguistic interpretation of NIS security depending on the incident rate and the severity of the incident consequences on the asset, and it builds an aggregate assessment and linguistic interpretation of the airport NIS risk for assets under consideration given their importance. The model allows the calculation of the amount of financial loss reserves in the implementation of risks considering the aggregated assessment and linguistic interpretation of the airport NIS security risks.
- The education expert model is also a useful tool for the education of future cyber criminologists in the aviation sector, which represents a new area of application of the results, and the designed software will expand upon the available digital aviation education tools, providing a modern approach to the training of aviation professionals.

The limitations to the study include:

- We created an educational expert model for the training of future aviation personnel in compliance with the principles of the teaching process, which apply to all aspects of teaching, i.e., the teaching activity of the teacher, teaching methods, and material means of teaching. We respected the basic didactic principles such as the principles of clarity, adequacy, durability, and systematics. We created a system in which each didactic principle is in a dialectical relationship with others. The main limiting factor is that the proposed educational model does not replace narrowly specialized software tools for teaching the technical security of information and information systems.
- The proposed educational expert model is focused on the comprehensive training of students in an area in which the experience of experts from real cases, real aviation practice, and real forensic investigation of technical failures, and especially the human factor in aviation, is significantly used. The proposed educational expert system is not intended to replace systems for technical assurance of reliability and safety of information systems in aviation. We aimed to contribute to the training of personnel, to handle such tasks with the support of modern technology.

In the future, we plan to focus on the creation of a new module entitled "Airport security and crisis management" for the Simulation Center with the support of the WebGIS platform, and functionalities for airport security management, airport crisis management, and airport network and information systems (NIS) security using the software created in this study. We will also focus on the development of the methodology for airport NIS, cybercrime, and cyber criminology experts (ANIS3C experts). We would like to research and support the development of education on cybercrime criminology in the aviation sector using the model of the perpetrator of the incident or cyber-attacks of the airport NIS, and evaluate the hybrid cybercrime prevention model for the airport NIS. We will include expert measurement and evaluation of cybercrime prevention in the context of airport security,

expert measurement, and the framework for the prevention of financial losses at the airport and for air transport.

These scientific outputs will form a valuable platform to promote cooperation between research centers, enabling the necessary sharing of research knowledge at the national and international levels in the transport sector and in the aviation sub-sector. This will support the building and strengthening of cyber capabilities and the creation of new mechanisms for ensuring cyberspace management, as well as improving risk management processes at airports and in the aviation sector for the benefit of the safety of all aviation activities with an emphasis on air transport. Lastly, the development of new methodologies and models is essential for the development of concepts of education systems in the field of aviation information security and air transport security and safety in Slovakia. It requires reforms aimed at updating specialized studies and training programs on this agenda. Experiences and results are applicable in other countries as well.

**Author Contributions:** Conceptualization, M.K.; methodology, M.K., V.P., and B.G.; data curation, S.S., R.A., and J.K.; investigation and formal analysis, M.K., V.P., B.G., R.A., S.S., W.Y., J.C., M.G., J.K., P.K., and M.A.; supervision and project administration, M.K.; writing—original draft, M.K., V.P., B.G., and R.A.; funding acquisition, S.S. All authors have read and agreed to the published version of the manuscript.

**Funding:** This research was supported and the APC was funded by the Slovak Research and Development Agency (the project No. APVV-17-0167—"Application of the Self-regulatory for the Preparation of Flight Crew").

**Conflicts of Interest:** The authors declare no conflict of interest.

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
