# Peer review of "Educational Model for Evaluation of Airport NIS Security for Safe and Sustainable Air Transport"

_sustainability, doi:10.3390/su12166352_

Round 1
Reviewer 1 Report
1. The abstract should include the numerical validations of the proposed work
2. The introduction is too lengthy
3. Proposed methodology was well written
4. The discussion part is so general and not so specific to the proposed work
5. The conclusion must be precise with illustrating the major outcomes, advantages and limitations of the proposed work in single paragraph
Author Response
We agree and respect the reviewers' comments on our scientific paper. Thanks to the reviewers for their advices.
The first reviewer's comments and suggestions to Authors:
- The abstract should include the numerical validations of the proposed work
We updated the abstract in the lines 26-29, and 36-48:
The paper aims to present the educational information model and software of the airport network and information systems risk assessment primarily intended for aviation education and training of professionals to fulfill such roles for ensuring safe and sustainable air transport.
Based on a fuzzy expert model, the selected scenario, and the input data established separately for airport assets by the group of 23 experts from aviation practice and university the following three assessments of airport network information system assets were obtained. Asset A_2 (Meteorological Information Systems) has an insignificant risk with an estimated R_2 = 0.1162. Assets A_1 (Air Traffic Control and Management (ATM), Navigational Aids and Approach) and A_3 (Runway Monitoring System) received a low risk of airport NIS security with ratings R_1 = 0.2623 and R_3 = 0.2915 respectively. An airport NIS risk assessment is aggregated Y = 0.2288, indicating a low degree of security risk to the airport's network and information systems. Aggregated risk assessment of airport NIS, including financial loss data, is Y (Z) = 0.1438, and this presents a low degree of security risk to the airport's network and information systems. Scenarios for evaluating airport assets are changing for students during education. The results of the developed model and its software will be part of the Simulation Center of the Faculty of Aeronautics.
- The introduction is too lengthy
We have shortened and updated the Introduction in the lines 64-70, 74-81, 94-103, 131-160.
- Proposed methodology was well written
- The discussion part is so general and not so specific to the proposed work
We shortened the Discussion section of paper in the lines 508-522.
We updated the Discussion section of paper in the lines 530-578:
The personnel within the air transport ecosystem work in different workplaces, in different work positions. Individual failure of the human factor, incident in work procedures, or failure of technology in one section of airport information systems may jeopardize the integrity and security of the entire air traffic support system. For this reason, the training of students is carried out in 3 levels, such as general aviation education, training of aviation specialists, and training using virtual training and information, simulation, and communication technologies in the aviation environment. The proposed model respects the division of students, future employees at airports into the following basic groups: safety and security, management / airline / airport operations, IT and computers in aviation, passenger management and services (air cargo and air passenger transport).
As part of general aviation education for the required aviation knowledge and skills of staff, students are prepared for the identification, analysis, assessment, and management of risks with an emphasis on:
Safety and Security: airport security, protection of persons and property, authentification systems, baggage screening systems, smart surveilance systems, customs and immigration service, in-line explosive detection systems (IEDs), perimeter intrusium detection systems (PIDS), emergeny response system;
Management / Airline / Airport operations: air traffic control and managment (ATM), navigational aiids and approcach, metorological information systems, airport operation planning (AOP), network operation plannig (NOP), dparture control systems (DCS), ground handling, de-icing systems, airfield lightning control systems, runway monitoring systems;
IT and computers in aviation: information systems for managers, local area network systems (LAN), communications systemss, radio spectrum management systems, hardware and software at the airport / in the air transport, wide area network (WAN), network security management in the aviation environment, flight information display system (FIDS), etc.
Passenger management and services (air cargo and air passenger transport): air cargo security procedures, passenger check-in and boarding, staff authentification system, etc.
The modular system of the proposed information model respects the mentioned areas of education and has functionalities for the preparation of students for aviation according to various scenarios and input data from the expert database, which are relevant for these areas in aviation. The proposed modular system forms the basis for the second level of education, the aviation specialists, in which its modules and functionalities are extended, for example, as „Airport security and crisis management“, decision support for airport security and crisis management for the protection of critical infrastructure entities in times of emergency such as COVID-19, for airport company, etc.
It is not possible to use all threat and risk scenarios in the paper within the air transport environment, which can be used in the aviation education of students. The proposed algorithm and created SW of the web application extend the tools of aviation education. The algorithm was verified on the example of evaluation of 3 selected airport network assets: A1 - Air Traffic Control and Management (ATM), Navigational Aids and Approach; A2 Meteorological Information Systems; A3 Runway Monitoring System according to the selected scenario and input data from 23 experts from aviation practice and aviation education.
An example from selected airport network assets allows students to build the ability to identify, describe, and analyze the threat, and risks, as well as numerically express the level of risk. The mentioned methodological procedure offers students as applicable to other areas and workplaces in aviation. The proposed model is another tool for realizing these phenomena, their complexity, and their causality. It helps to gain knowledge in the field, awareness of the importance of the human factor in all processes relevant to air traffic and air transport. The proposed model makes available in an interesting way for the young generation the issue of aviation, which has the potential to cause material and financial damage, even the loss of human lives.
We shortened the Discussion section of paper in the lines 579-606.
We updated the sentence in the lines 607-609:
Sharing research knowledge directly linked to real-world testing and with a focus on creating a predictable platform in the field of cybersecurity education will support the lack of public sector cooperation with the private and academic sectors so far.
- The conclusion must be precise with illustrating the major outcomes, advantages and limitations of the proposed work in single paragraph
We updated the Conclusion section of paper in the lines 629-676:
The research of the actual task of developing an educational expert model of airport NIS risk and security incidents measuring and assessment in the framework of aviation sector was conducted with the following major outcomes, advantages and limitations of the proposed work:
Major outcomes
- Creation of the educational expert model based on the fuzzy logic (the modular system of the proposed educational information model) for obtaining an aggregate airport NIS risk assessment is presented in the form of an 8-step knowledge-based algorithm, using linguistic variables, which reveals the inaccuracy of the input data, for preventing cybercrime and producing the outputs: risk assessment and linguistic interpretation of the airport NIS safety, based on the "incident rate" and the severity of the effects of the incident on the asset; aggregated assessment and linguistic interpretation of the airport NIS risk for assets under consideration with concerning their importance; aggregated assessment and linguistic interpretation of airport NIS security risk taking into account financial losses;
- Based on the developed educational expert model the innovative software used on the Simulation Center of the Faculty of Aeronautics at the Technical University in Košice was constructed for the multi and interdisciplinary training of aviation specialists in this agenda. The algorithm of the education information model was presented and verified on the example of airport NIS security risk measuring and assessment on 8 personal data security threats and 3 airport NIS assets;
Advantages
- It increases the objectivity of expert assessments, using the input linguistic variables of consequences of the realization of security threats of personal information of airport NIS and the degree of possibility of such sales; it reveals uncertainties of fuzzy expert judgment through the function "incident rate"; it performs the risk assessment and linguistic interpretation of NIS security, depending on the "incident rate" and the severity of the incident consequences on the asset; it builds an aggregate assessment and linguistic interpretation of the airport NIS risk for assets under consideration given their importance; the model allows to calculate the number of financial loss reserves in the implementation of risks and taking into account the aggregated assessment and linguistic interpretation of the airport NIS security risks.
- An education expert model is also a useful tool for the education of future cyber criminologists in the aviation sector, which represents a new area of application of the results, and the designed software will expand the tools of digital aviation education and a modern approach to the training of aviation professionals as was mentioned;
Limitations
- We have created an educational expert model for the training of future aviation personnel in compliance with the principles of didactics in the teaching process, which apply to all aspects of teaching, ie the teaching activity of the teacher, teaching methods or material means of teaching. We respect the basic didactic principles such as the principle of clarity, adequacy, durability and systematics. They create a system in which each didactic principle is in a dialectical relationship with others. The main limiting factor is that the proposed educational model does not replace narrowly specialized software tools for technical security of information and information systems.
- The proposed educational expert model is focused on the comprehensive training of students in an area in which the experience of experts from the real cases, real aviation practice and real forensic investigation of technical means failures, but especially the human factor in aviation, is significantly used. The proposed educational expert system has no ambition to replace systems for technical assurance of reliability and safety of information systems in aviation. It is to contribute to the training of personnel, to handle such tasks with the support of modern technology.
We shortened the Conclusion section in the lines 678-724.

Reviewer 2 Report
The paper presents the expert model for evaluation of airport system security to support the security Management for the education in air transport area in Slovakia.
First of all it's a little strange that the model and system is made for country that doesn't have its own national airlines company. But maybe it will be in a near future.
Next point is the use of fuzzy logic for such important issue as information security, information system secuirty, especially in the air transport area. I'm really afraid of the results of such systems and their reliability, assurance and safety.
Another weak points of the paper are as follows:
- the captions of the figures and tables are too general and not clear
- the names of the section and subsection are too general
- there are many editing errors, typoes and stylistic errors
- the screenshots of the system/application (it looks rather as an applocation, not a system) are very general.
Author Response
We agree and respect the reviewers' comments on our scientific paper. Thanks to the reviewers for their advices.
Regarding the comments and suggestions of the second reviewer:
The paper presents the expert model for evaluation of airport system security to support the security Management for the education in air transport area in Slovakia.
First of all it's a little strange that the model and system is made for country that doesn't have its own national airlines company. But maybe it will be in a near future.
We are the small country and we have no any national airliner. The aviation world is not just made up of airlines. At present, our Faculty of Aviation has 5,000 graduates who work in various positions around the world. Aviation education, and specially innovated digital aviation education tools, are of great importance for the training of new aviation personnel.
Next point is the use of fuzzy logic for such important issue as information security, information system secuirty, especially in the air transport area. I'm really afraid of the results of such systems and their reliability, assurance and safety.
As we mentioned in the Conclusion section of paper. The proposed educational expert model is focused on the comprehensive training of students in an area in which the experience of experts from the real cases, real aviation practice and real forensic investigation of technical means failures, but especially the human factor in aviation, is significantly used. The proposed educational expert system has no ambition to replace systems for technical assurance of reliability and safety of information systems in aviation. It is to contribute to the training of personnel, to handle such tasks with the support of modern technology.
Another weak points of the paper are as follows:
- the captions of the figures and tables are too general and not clear
We changed the title of Figure 1 in line 245 to the title:
Figure 1. Structural Diagram of the Airport Assets Risk Assessment Expert Model.
We changed the title of Section 2 in the lines 253-254 to the title:
Model of experts’ input data for the selected scenario of airport assets assessment
We changed the title of Table 1 in the line 274 to the title:
Table 1. Airport NIS assets - the severity of the effects of the asset incident
We changed the title of Table 1 in the line 312 to the title:
Table 2. Experts’ Input data for airport assets threats assessment
We changed the title of in the lines 344-349 to the title:
Figure 2. Fuzzification of the experts ‘ input data for the selected scenario of airport assets assessment
M – Minimum consequence of the threat, interval value on the percentage scale [0; 20]
A – Average consequence of the threat, interval value on the percentage scale [20; 50]
H – High consequence of the threat, interval value on the percentage scale [50; 80]
C – Critical consequence of the threat, interval value on the percentage scale [80; 100]
We changed the title of Table 3 in the line 431 to the title:
Table 3. Input data, expert assessment for the selected scenario of airport asset .
We changed the title of Table 4 in the line 432 to the title:
Table 4. Input data, expert assessment for the selected scenario of airport asset .
We changed the title of Table 5 in the line 433 to the title:
Table 5. Input data, expert assessment for the selected scenario of airport asset .
We changed the title of Table 6 in the lines 440-441 to the title:
Table 6. Fuzzification of the experts ‘ input data for the selected scenario of airport assets assessment
We changed the title of Table 8 in the lines 471 to the title:
Table 8. Calculation of incident financial loss for the selected scenario of airport assets assessment
We updated the sentence in the lines 485-486:
Aggregated risk assessment of airport NIS, including financial loss data, is0.1438, and this presents a low degree of security risk to the airport's network and information systems.
We changed the title of Figure 3 in the lines 493-495 to the title:
Figure 3. Software window for the modular system of the selected airport assets assessment according to the students training scenario – A1 Air Traffic Control and Management
We changed the title of Figure 4 in the lines 493-495 to the title:
Figure 4. Software settings for consequences, linguistic and numerical security risk level of airport’s network information system assessment according to the students training scenario
- the names of the section and subsection are too general
We changed the title of Section 2 in the lines 253-254 to the title:
Model of experts’ input data for the selected scenario of airport assets assessment
We changed the title of Section 3 in the lines 422-423 to the title:
- Results of education fuzzy expert model verification according to the selected student training scenario
- there are many editing errors, typoes and stylistic errors
As part of the revision and preparation of the article, we did an English editing service MDPI (ID english-20572, 15 July 2020).
- the screenshots of the system/application (it looks rather as an applocation, not a system) are very general.
The modular system of the proposed educational information model respects the mentioned areas of education and has functionalities for the preparation of students for aviation according to various scenarios and input data from the expert database, which are relevant for these areas in aviation. The proposed algorithm of the educational expert model was used for the creation of a web application, as a modular system for training personnel for various areas of airport network assets, with an emphasis on managing such tasks in an environment of uncertainty and risks of air traffic.

Round 2
Reviewer 2 Report
The weak points to improve of the paper:
- Introduction is too long. Most content of subsection 1.1 and 1.2 should be move to the new sections, just after the Introduction. E.g. I don’t understand why the authors put very important, from the point of view of their work, figure in introduction?
- What are these variables (?) “Ks” in section 2? Why they are so important to describe them in some details? It is not clear, it is not clearly explained.
- Training scenarios/cases should be more described
- The working results of the created software could be shown, described and analyzed.
Author Response
We agree and respect the reviewers' comments on our scientific paper. Thanks to the reviewers for their advices.
The reviewer No. 2, Round 2: Comments and suggestions to Authors:
- Introduction is too long. Most content of subsection 1.1 and 1.2 should be move to the new sections, just after the Introduction. E.g. I don’t understand why the authors put very important, from the point of view of their work, figure in introduction?
We updated the section 2 in the lines 162-201:
The introduction has been shortened and modified. Content of subsection 1.1. Overview of Domestic and Foreign Research Studies remained to support the relevant references to the issue in the Introduction. The content of subsection 1.2 became part of the following section 2 Formal problem statement and Model of Input Data, including Figure 1.
- What are these variables (?) “Ks” in section 2? Why they are so important to describe them in some details? It is not clear, it is not clearly explained.
In lines 163-170 we mentioned:
In the subsection 2.1 we described that we have a set of information assets of civil aviation security [2] A={A1;A2;...;An}, for which many threats to the security of personal data of network and information systems have been identified (the airport network threats in real-time, malicious or anomalous patterns, airport security threats, air transport security and safety threats, aviation communication systems threats, air traffic control threats, the threat of fraudulent acquisition and use of air passengers' private identification information, airport health threats including antibacterial and antiviral protection, etc.) K={Ki1;Ki2;...;Kim}.
Every security threat Kij for an asset Ai, is assessed by a group of experts in the form of the input data (Tij;μij).
In lines 258-268 we have added an explanation
For educational purposes, we have the airport facilities divided into 4 main areas, 25 subsystems of airport facilities (A1-A25), for which we can use (simulate) variable threats (Kii) to the security of personal data in airport network and information systems, etc., which have the potential to have serious consequences. Our students should consider (identify, analyze and evaluate) security threats to the airport NIS implemented using inter-network protocols for selected airport facilities, according to the situation formulated by their training instructors. The selected situation for the training of students determines the training scenario for specific lessons. The situation is classified according to the subjective judgments of experts (managers). We have a set of information assets of civil aviation security for which many threats to the security of network and information systems, etc., have been identified that why they are so important to describe them to our students in some details.
- Training scenarios/cases should be more described
Lines 382-391
Training scenarios are created by training instructors from 4 main areas of airport facilities, and their 25 subsystems. The number of areas and subsystems creates space for scenario variations. Specific scenarios are created first within one area and its several subsystems. Subsequently, the situation for students' decision-making is complicated by the creation of a scenario from at least 2 areas of airport facilities and 4 of their subsystems, etc. Each training scenario pursues an educational goal so that students acquire knowledge and skills in identifying, analyzing, and assessing risks in selected areas and subsystems of airport facilities, in creating solutions (strategies) and especially in risk management. Training scenarios are aimed at identifying and understanding the causes of failure of the human factor, the technical system, or a combination of these causes (resources).
- The working results of the created software could be shown, described and analyzed
Lines 477-496
Figure 5 Screenshot of the final assessment of selected airport assets NIS risk for support of decision-makers (students)
The working results of the created software obtained:
- Setting up of 25 airport NIS assets and 8 personal data security threats to assess the airport NIS security risks. The input data are presented in the form of linguistic assessment (term-set of 4 linguistic variables) of consequences of the realization of security threats of personal data of NIS of the airport and degree of potential threat occurrence;
- An approach for calculating the number of financial loss reserves for the realization of airport NIS risk is presented and a function for interpreting the dependence of financial loss reserves is created;
- An education model for obtaining an aggregate airport NIS risk assessment is presented in the form of an 8-step knowledge-based algorithm, using linguistic variables, which reveals the inaccuracy of the input data, for preventing cybercrime and producing the outputs: risk assessment and linguistic interpretation of the airport NIS safety, based on the "incident rate" and the severity of the effects of the incident on the asset; aggregated assessment and linguistic interpretation of the airport NIS risk for assets under consideration with concerning their importance; aggregated assessment and linguistic interpretation of airport NIS security risk taking into account financial losses.

Round 3
Reviewer 2 Report
I propose to accept the paper in the current form.
This manuscript is a resubmission of an earlier submission. The following is a list of the peer review reports and author responses from that submission.
Round 1
Reviewer 1 Report
Major comments
- In the introduction it is not made clear what the focus of the proposed methodology is. Is it national (Slovakia?) or general? If general, then why the focus on Slovakian resources in the introduction? If Slovakian, then what are the specific problems Slovakia faces?
- Traditionally, experts are incorporated in risk assessment strategies, but it is not reflected in the paper. Section 1.2 mentions that experts are incorporated in valuation of different aspects of security risks, but it is not reflected that this is common practice. See for instance the ISO 31000:2018 risk management guideline, and Washington, A. S. M. E. "All-hazards risk and resilience: Prioritizing critical infrastructures using the RAMCAP Plus [hoch] SM approach." ASME, 2009. It should be clearly reflected what is novel in the present approach, and which problem it solves in existing methods. This is currently lacking. Figure 1 reflects a more or less standard order of events for the assessment of risks.
- Later in the paper the emphasis is more and more on fuzzy logic. However, this idea of using fuzzy logic in security models is not new. See for instance Skorupski, Jacek, and Piotr Uchroński. "A fuzzy model for evaluating airport security screeners' work." Journal of Air Transport Management48 (2015): 42-51. It is important to show what is novel in the current approach, and this is currently not clear.
- As expert input is an important factor in this work, it is important to specify, for each value that is shown in the tables, what the source is. Currently, only a general statement about data sources is provided. This makes it hard to judge the quality of the data that is provided.
Minor comments
Line 50: National Information Security Strategies of which country? Slovakia?
Line 266 onwards: In the description of the method you consider threats for different assets, but here you only mention threats (i.e. kij vs ki).
Line 315: why did we take this S-like membership function?
Line 352: marginal means ‘minor and not important; not central.’, while it is the most important risk.
Line 409: “let's do it step by step.” -> too informal.
Reviewer 2 Report
Strengths:
- There’s a sound formalistic study approach to a topic that is fundamentally of importance. Using fuzzy expert estimates makes sense in the presented context and provides a sensible range of estimates that can be used for security measures.
- The stepwise directions are clear and helpful for anyone who wants to conduct a similar exercise using this fuzzy expert model framework.
- Nice to integrate modelling of financial losses, too. Ultimately, this is what it comes down to for most of the considered attack vectors - but not all!
Weaknesses:
- I am not sure this journal is a sensible fit. Maybe I define sustainability differently, but even in the transport subsection I could not find "security" among the relevant topics. An aviation or computer security venue would certainly provide a better fit.
- This largely reads like a consulting exercise for a specific airport, rather than a scientific venture with a hypothesis. In large part is is too detailed and specific to the use case at hand, for example the results should be presented in a better fashion. Plugging all the individual numbers into formulas is not very appropriate for a research paper (at most in an appendix, maybe). Applying an existing method in a specific context may still be original research, but it should provide more insights than just a recipe-style application. What are the alternatives? How does this method compare to others and on which metrics?
- I don’t see considerations for loss of life, in particular for attacks on operational aircraft/ATC systems. If we think about sustainability, safety should definitely be at the forefront.
- We here nothing about the experts that are used for this study, demographics, selection etc. Where should one get them from, what qualifications are needed?
Details:
- The abstract should make it more clear with concrete details and outcomes what the (expert) reader should expect from the article. What problem do you solve, how, and what are the results? Just saying that you developed a tool is very unhelpful.
- I would suggest to cut and re-write the introduction. It should focus more on the reader takeaways than it does right now. What existing problem do you actually want to solve (better) with this paper? Everyone reading this knows that cybersecurity is important. It is very tedious to read about all the organisations saying so, and very unsurprising.
- National Information Security Strategies - which ones / which countries?
- Part of the language is a bit convoluted and sentences occasionally need to be re-read several times to be understood.
- It is not clear how/why you focus on cyber crime specifically. Lack of cybersecurity can invite many actors with different skills and motivations. Is there much precedence? Is it different from other sectors? The software tools that you list are very generic.
- What is sector specific here? There is a lot of literature on the aviation communication systems (ADS-B, ACARS, ILS etc.), it would be more interesting to integrate these into this model of expert opinion. As you say, "Security experts tend to respond only to security risks they understand." and this does not really seem to be fixed in your approach.
- The related work section mixes in a discussion of methodology (from “Our study” onwards). This should be separate.
Nitpicks:
- A few typos throughout: e.g. "sector analyzes” -> analyses